# eCommerceGAN: A Generative Adversarial Network for e-Commerce

**Ashutosh Kumar**[*]
Indian Institute of Science
Bangalore, India
ashutosh@iisc.ac.in

**Arijit Biswas & Subhajit Sanyal**
Amazon India Machine Learning
Bangalore, India
{barijit,subhajs}@amazon.com

## Abstract

E-commerce companies such as Amazon, Alibaba and Flipkart process billions of orders every year. However, these orders represent only a small fraction of all plausible orders. Exploring the space of all plausible orders could help us better understand the relationships between the various entities in an e-commerce ecosystem, namely the customers and the products they purchase. In this paper, we propose a Generative Adversarial Network (GAN) for e-commerce orders. Our contributions include: (a) creating a dense and low-dimensional representation of e-commerce orders, (b) train an ecommerceGAN (ecGAN) with real orders to show the feasibility of the proposed paradigm, and (c) train an ecommerce-conditional-GAN (ec$^2$GAN) to generate the plausible orders involving a particular product. We evaluate ecGAN qualitatively to demonstrate its effectiveness. The ec$^2$GAN is used for various kinds of characterization of possible orders involving cold-start products.

## 1 Introduction

Major e-commerce companies such as Amazon, Alibaba, Flipkart and eBay have billions of products in their inventories. However, the space of products is of much lower dimensions as there is an inherent underlying structure imposed by product categories, sub-categories, price ranges, brands, manufacturers, etc. Similarly on the customer side, though there are several hundreds of millions of customers, they reside in a lower dimensional space as customers can be grouped together based on similarity of purchase behaviors, price/brand sensitivity, ethnicity, etc. Thus the space of e-commerce orders is an interaction of samples from these two spaces where certain interactions are plausible while others are unlikely to happen ever. For example, it is unlikely for a user, who "belongs" to a group of users who are jazz aficionados, to purchase a heavy metal album. However, the real orders, i.e., the orders which have been placed in an e-commerce website represent only a tiny fraction of all plausible orders. Exploring the space of all plausible orders could provide important insights into product demands, customer preferences, price estimation, seasonal variations etc., which, if taken into consideration, could directly or indirectly impact revenue and customer satisfaction.

In this paper, we propose an approach to learn the distribution of plausible orders using a variation of Generative Adversarial Network (Goodfellow et al., 2014). The three major contributions of this paper are: (i) **Order Representation:** We represent each order as a tuple of {customer, product, price, date} and propose a novel order embedding approach. (ii) **ecommerceGAN (ecGAN):** We use a Wasserstein GAN (WGAN) (Arjovsky et al., 2017) to train a generator which can generate plausible e-commerce orders. We train the ecGAN to explore the overall viability of the proposed approach. (iii) **ecommerce-conditional-GAN (ec$^2$GAN):** We also propose a conditional GAN to generate orders which are conditioned on a particular product, i.e., its representation. In the e-commerce space, it is desirable to understand the characteristics of future orders involving a particular product, especially when a new product is launched. As an example, being able to estimate the demographics of the potential customer-base for a new product can help companies make more informed decisions about their marketing efforts. If we can predict the gender, age, tenure and location of possible future customers of a cold-start product accurately, they can be targeted with personalized deals and recommendations. The proposed ec$^2$GAN takes a product representation as input (along with random

---

[*]Work done as an intern at Amazon India Machine Learning, Bangalore

noise vectors) and generates many plausible orders involving this particular product. The generated orders are analyzed to characterize the possible customers and determine seasonal demand. To the best of author's knowledge this is the first work that applies GANs in the e-commerce domain to learn a generator for plausible e-commerce orders.

## 2 METHOD

An order is defined in terms of the following key elements: {customer, product, price, date}. We use an unsupervised approach to represent each product in an e-commerce ecosystem using a dense and low-dimensional representation. The word2vec representations of all the words in a product title are (weighted-) averaged to create 128-dimensional dense product embeddings. The customer representations are obtained using a Discriminative Multi-task Recurrent Neural Network (RNN), where different signals pertaining to a customer's recent purchase history are explicitly encoded into her embedded representation (128-dimensional). The price of the product is represented using a log-transformed and normalized value (1-dimensional). Date of purchase of an order is represented as a 7-dimensional vector, which captures both the circularity of months/days and the difference between the current date and a pre-decided epoch. The individual element representations are concatenated to create a 264-dimensional order representation. Please refer to Appendix-5 for more details.

In ecGAN, we use a WGAN to model the space of order representations. Let us assume that an order $\mathbf{O}_n \in \Phi$ is a tuple $\{\mathbf{C}_i, \mathbf{P}_j, p_k, \mathbf{D}_l\}$, where $\mathbf{C}_i$ is the representation of the i[th] customer at that point in time when she makes this order/purchase, $\mathbf{P}_j$ is the product representation of the j[th] product, $p_k$ is the price of the product and $\mathbf{D}_l$ represents the vector corresponding to the date on which the order was made. The discriminator in ecGAN is fed with the set of orders $\Phi$ and it tries to discriminate them from the order set $\tilde{\Phi}$, which is generated by the ecGAN generator.

The generator, which is a fully connected network with two hidden layers and ReLU (Nair & Hinton, 2010) non-linearity at the end of each hidden layer, maps the noise vectors ($z$) to feasible orders ($\tilde{\mathbf{O}}_n$). The 264 dimensional fake orders ($\tilde{\mathbf{O}}_n$) from the generator are compared with the 264 dimensional real orders ($\mathbf{O}_n$) in the discriminator. The discriminator, which is also a fully connected network with two hidden layers and ReLU non-linearity at each hidden layer, maps the order inputs to real or fake labels.

In ec$^2$GAN, we want the generator to generate orders which are conditioned on a particular product. We call the modified version ecommerce-conditional-gan or ec$^2$GAN. To this end, we make a couple of key modifications to the basic ecGAN architecture:

- In the generator, we feed a vector $z'$, which is the concatenation of random noise vector $z$ and the product representation $\mathbf{P}_j$, i.e., $z' = [z, \mathbf{P}_j]$. This ensures that during test time, when we want to generate a set of feasible orders, we can condition the network on any product of our choice.
- We also add a reconstruction loss component to the generative loss. The reconstruction loss, denoted as $J^{(R)}$, enforces the product components of the generated orders to be same as the product representation which we conditioned on. We define the reconstruction loss as the Euclidean distance between the actual product vector $\mathbf{P}_j$, which is the input to the generator, and the generated product vector $\tilde{\mathbf{P}}_j$, i.e., $J^{(R)} = ||\mathbf{P}_j - \tilde{\mathbf{P}}_j||$. The new generator loss in the ec$^2$GAN is defined as $\alpha J_W^{(G)} + (1 - \alpha)J^{(R)}$, where $J_W^{(G)}$ is the generator loss corresponding to the WGAN generator and $\alpha$ is a tunable parameter.

## 3 EXPERIMENTS

In this work, we use the products from the apparel category for model training and evaluation. We randomly choose 5 million orders made over the last one year in Amazon to train the proposed models. Once we obtain the {customer, product, price, date} tuple corresponding to each order, we get their corresponding 264-dimensional representations using the proposed order representation approach and use them to train our models.

**ecGAN Qualitative Analysis:** We randomly choose 10K real orders and generate 10K orders using ecGAN. We mix all the data and perform t-SNE (van der Maaten & Hinton, 2008) and project them to a 3-dimensional space. The projected space is explored from various perspectives and are shown in Figure 1. We make three key observations: (a) the real orders and the generated orders are mixed well with each other throughout the projected space (e.g., region A, B and C in Figure 1a), (b) the generator has learned to generate data from all the modes in the real order distribution (e.g., region D,

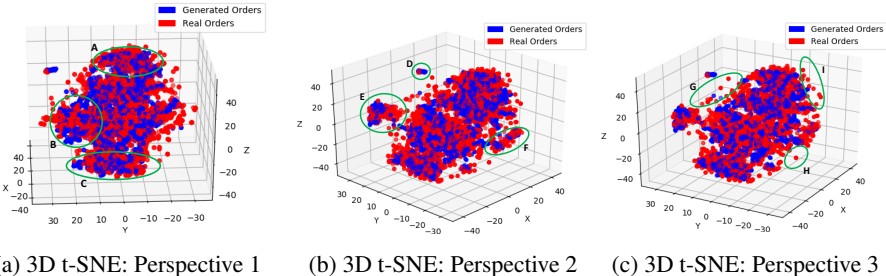

(a) 3D t-SNE: Perspective 1      (b) 3D t-SNE: Perspective 2      (c) 3D t-SNE: Perspective 3

Figure 1: Different perspective of *orders* in a projected 3-D space (t-SNE). Blue represents generated *orders*, whereas red denotes real *orders* (best viewed in electronic copy).

E and F in Figure 1b) and (c) the generator has learned to ignore the outliers or unlikely orders in the real order-set (e.g., region G, H and I in Figure 1c). Overall, ecGAN is able to generate orders that are quite similar to the real order distribution.

**ec$^2$GAN Distribution Comparison**: The proposed ec$^2$GAN can be used to generate possible future orders involving a new e-commerce product. We generate orders corresponding to a few products (see Appendix-6 for details) and compare them with the ground-truth future real orders based on customer characteristics and seasonal demands. The normalized gender, tenure, purchase volume (for customers) and seasonal demand distribution of the generated and the ground-truth orders are shown in Figure 2. We observe that in almost all of these scenarios, the proposed ec$^2$GAN can effectively emulate the ground-truth distribution. For example, given a product with title "Women Party Dress", the generated orders indicate that it will be purchased by female customers 73% of the times, whereas male customers could buy this product 27% times [1]. Similarly, ec$^2$GAN predicts that "Men Shorts" will be purchased during summer months 63% of the times, whereas "Women Winter Shawl" will be purchased 67% of the times during winter. Although it is often possible to determine the gender or season from the details of a product, the tenure or purchase volume of the prospective customers are not immediately inferable from the product details. In such scenarios, generating the possible orders of a product using ec$^2$GAN seems to be a natural way for order characterization. From Figure 2, it is clear that ec$^2$GAN has been able to mimic the real purchase volume and tenure data distribution in various products such as "Boys Hoodie", "Women inner wear", "Baby onesie" and "Men Lounge Pants". These results indicate the effectiveness of the proposed ec$^2$GAN. We also note that all these analysis would have been fraught with inaccuracies had we been plagued with issues of mode-collapse. The fact that the characteristics measured from the generated samples closely mimic the characteristics of the real samples, indicates the efficacy of our trained model.

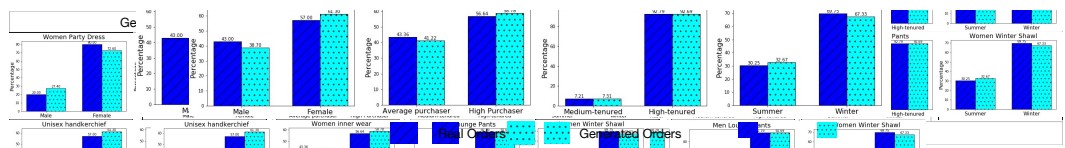

Figure 2: Comparison between ec$^2$GAN and ground-truth. The plots are normalized, i.e., if x denotes % of winter purchases and y denotes % of summer purchases, we plot $\frac{x}{x+y}$ and $\frac{y}{x+y}$ respectively.

We perform a comprehensive set of evaluations for ecGAN and ec$^2$GAN in Appendix-7.

## 4   CONCLUSION AND FUTURE WORK

In this paper, we propose two novel variations of Generative Adversarial Networks for e-commerce: ecGAN and ec$^2$GAN. The proposed ecGAN learns to generate orders which are similar to the real e-commerce orders. The proposed ec$^2$GAN is applied to generate orders corresponding to a new e-commerce product. Preliminary qualitative and quantitative results are presented to evaluate the effectiveness of the proposed methods. We believe that ec$^2$GAN will be useful for various practical e-commerce problems such as product recommendation and stock-keeping (please see Appendix-6). The proposed techniques should also inspire successful GAN application and evaluation in various other domains such as healthcare, transportation, finance and sports.

---

[1]A female oriented product is typically bought by males for gifts or family members.

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

## 5 APPENDIX - ORDER EMBEDDING

An e-commerce order is defined in terms of the following key elements: {customer, product, price, date}. We compute feature representations for each of these individual elements and concatenate them to obtain a dense, low-dimensional representation for the order. This order representation is eventually fed to the proposed ecGAN and ec$^2$GAN.

### 5.1 PRODUCT EMBEDDINGS

We use an unsupervised approach to represent each product in an e-commerce ecosystem using a dense and low-dimensional representation. The word2vec representations of all the words in a product title are (weighted-) averaged to create dense product embeddings. First, we train a word2vec (Mikolov et al., 2013) model using a corpus of titles and descriptions pertaining to 143 million randomly selected products. We obtain a vocabulary of around 1.4 million words along with their word2vec representations. Next, the same word2vec corpus is used to find the inverse document frequency (IDF) of all the words in the vocabulary. The word2vec representations corresponding to each word are multiplied with the corresponding IDF weights and are summed across all the words in the title. The representations are normalized by the total IDF score of all the words in the title. This approach brings similar products close to each other in the learned representation space. The embeddings created using this method are 128 dimensional and the values range between -1 and 1.

## 5.2 CUSTOMER EMBEDDINGS

To represent each customer in an e-commerce system, we train a Discriminative Multi-task Recurrent Neural Network (RNN), where different signals pertaining to a customer's recent purchase history are explicitly encoded into her embedded representation. Each signal is captured by formulating a multi-class classification task. We use three different training tasks: (i) Predicting the next product group (such as clothes, food, furniture, baby-products etc.) purchased by a customer, (ii) Predicting how much price a customer will pay on the next purchase and (iii) Predicting after how many days the customer will purchase an item from the e-commerce company[2]. The proposed architecture contains an RNN with LSTM cells as the input layer(motivated by (Zolna & Romanski, 2017)) which takes the sequence of products i.e., their representations as described in Section 5.1 purchased by a customer and creates a hidden representation. The hidden representation, which we refer to as "customer embedding", is fed into multiple classification units corresponding to the training tasks. The network is jointly trained with all the tasks in an end-to-end manner using alternating optimization. At each iteration of training, we randomly choose one of the tasks and optimize the network with respect to the loss corresponding to that task only. This approach brings two customers close in the semantic space if they have similar purchase history. Using this approach we represent each customer using a 128 dimensional dense and low-dimensional vector. The range of each feature is between -1 and 1.

## 5.3 PRICE

The prices of the products sold in an e-commerce system varies between a few dollars to tens of thousands of dollars. We first take logarithm of the prices to squash the effect of the large price values. We further normalize each log-transformed price between the range of -1 and 1.

## 5.4 DATE OF PURCHASE

Date of purchase of an order is represented as a 7-dimensional vector. The first component captures the difference between the current date and a pre-decided epoch. The next two components represent day of the month, followed by two components representing the day of the week and the last two components representing the month. The features have been carefully crafted to contain the information about circularity of days, i.e., Monday would be equally close to Wednesday as it would be to Saturday. We achieve this by projecting the possible days/months on the periphery of a unit circle and representing them using their sine and cosine components. Each of the features are normalized between -1 and 1.

We concatenate the 128 dimensional customer vector, 128 dimensional product vector, 1 dimensional price and 7 dimensional purchase date vector to obtain a 264 dimensional vector corresponding to each order. These vectors are dense, low-dimensional and semantically meaningful.

## 6 APPENDIX - APPLICATION PIPELINE

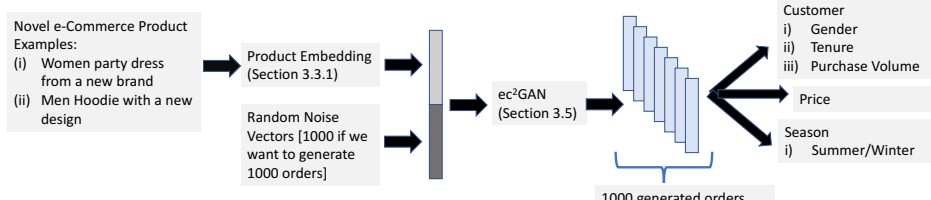

Figure 3: The Application Pipeline for ec[2]GAN.

In this section, we describe how the proposed ec[2]GAN can be applied to a novel e-commerce product in real-world scenarios, which in turn could potentially impact practical e-commerce problems such

---

[2]We tried various training task combinations for a fixed embedding dimension and found this subset to be most effective on various customer classification tasks.

as recommendation and stock-keeping. The application pipeline is provided in Figure 3. The steps are described below:

1. Obtain an embedding for a product using the proposed approach in Section 2.
2. The product embedding is concatenated with random noise vectors and fed as input to the ec$^2$GAN network. Eg. Suppose we want to generate $N$ orders(1000 in this paper), we sample $N$ different random noise vectors, concatenate each of them with the product embedding and provided them as input to the ec$^2$GAN. The ec$^2$GAN outputs $N$ different orders.
3. The generated orders are classified to determine the customer preference (gender, tenure and purchase volume), price and seasonal demands of a product.
4. **Recommendation:** Once we know the customer preference for a particular product, we could target the corresponding customers with that particular product. For example, if we know that a "a winter jacket from brand X and design Y" will be purchased a lot by middle-aged women, all the corresponding customers can be recommended this "winter jacket".
5. **Stock-keeping:** Similarly, once we know that "a particular handbag from brand X, color Y and material Z" will be sold more during the winter months, the warehouses could keep larger stocks of that item during the winter months.

## 7 APPENDIX - EVALUATION

### 7.1 ECGAN QUALITATIVE ANALYSIS(CONTD.)

- **Feature Correlation:** In this study, we compute the feature correlation between the real orders and the generated orders. Each order comprises 264 features. We randomly select three features $\{f_1, f_2, f_3\}$ from the feature set. We check if the correlation coefficient between $f_1$ and $f_2$ is more than the correlation coefficient between feature $f_1$ and $f_3$. For each triplet, we check if the feature relationship in the real data agrees with the feature relationship in the generated data. We randomly choose 100K triplets and determine the fraction of the times they agree. We observe that 77% of the times, the generated data match with the real data (whereas the baseline will be 50%). This also demonstrates that the generator has learned to generate orders which are similar to the real orders to an appreciable extent.

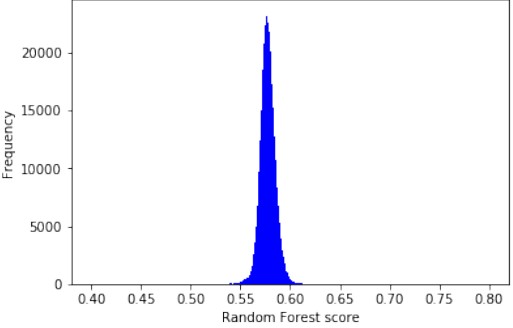

Figure 4: Histogram of data distribution scores in random forest leaves.

- **Data distribution in Random Forest Leaves:** We propose an indirect method to estimate the similarity between the real orders and the fake orders. We train a random forest classifier and look at the distribution of the real and fake orders at each leaf node of the forest. We randomly sampled 1 million real orders, generated 1 million fake orders using ecGAN and mixed them to create a set of 2 million orders. We label them as class 1. Next, we randomly permuted the columns of these orders and generate a random dataset of 2 million orders. This dataset is labeled as class 2. We train a binary classification model to separate class 1 from class 2. The random forest has 100 trees with maximum depth of 5 (32 leaf nodes in each tree, i.e., 32 clusters of the input data). Next, we trace each of the 1 million real orders and 1 million fake orders through each of the 100 trees and note down the nodes they traveled through. We compute for each leaf in each tree the "real-order-ratio", which is defined as the fraction of the real orders out of all the orders that end up in that node. Then for each order, we average the "real-order-ratio" of the leaf nodes they end

up in across all the 100 trees. If in each node, the real data and fake data are mixed up well, these scores should be close to 0.5 and should indicate that the generated orders are similar to the actual orders. We plot the histogram of the scores of all the real orders and plot them in Figure 4. This distribution (centered around 0.57) establishes that the real data and fake data are quite inseparable.

## 7.2 EC$^2$GAN QUANTITATIVE ANALYSIS

As discussed in the earlier sections, GAN evaluation is difficult. We propose an approach to quantitatively evaluate GANs. The evaluation technique is designed with e-commerce in mind but should be applicable to any other domain such as finance, transportation, health-care and sports. In ec$^2$GAN, a product representation is fed along with the noise vector to the generator and orders which involve the input product are generated. When a new product is introduced into an e-commerce system, it is extremely hard to determine the set of customers who would be interested in this product, at how much price the product will be sold and when the product will have high demand. Robust forecasting of these would enable e-commerce companies to better manage their inventories and adopt proactive supply chain optimizations. In this section, we discuss given a brand new e-commerce product, how we can characterize various components (customer demographics, price, date or seasonality) of the generated e-commerce orders. We propose an evaluation metric called Relative Similarity Measure or RSM which lets us quantitatively compare the generated orders and the real orders with respect to each characteristic. We note that the goal of this paper is not to produce state-of-the-art results in customer characterization, price estimation or demand prediction, where highly specialized solutions can be engineered for each of these problems independently. We apply ec$^2$GAN to these problems to demonstrate a quantitative way of evaluation of the proposed GAN technique, which is difficult to do otherwise. We compare ec$^2$GAN with a baseline order generation approach based on Conditional Variational Autoencoder (C-VAE) (Walker et al., 2016) and describe the details of the baseline approach in Section 7.2.1. The data which are used in the following evaluation experiments are completely disjoint from the dataset, which we used to train our model.

### 7.2.1 BASELINE ORDER GENERATION APPROACH

We compare the proposed approach with a baseline approach that uses Conditional Variational Autoencoders (Walker et al., 2016) for order generation. This approach is an extension of Variational Autoencoders (Kingma & Welling, 2013). However, the generation process can be conditioned on various kinds of signals such as labels, images etc., such that the generated samples are restricted by the input signal. For example, in (Walker et al., 2016), the authors propose to predict the future such as what will move in a scene, where it will travel, and how it will deform given any static image. In this work, we use the C-VAE for order generation given a particular product. The same dataset is used for training both the proposed ec$^2$GAN and the baseline C-VAE.

### 7.2.2 RELATIVE SIMILARITY MEASURE (RSM)

We propose the RSM to measure the relative similarity between the generated orders and the real orders. Let us assume that each product $p_i$ can be assigned a propensity score of $s_i^t$ by the ground-truth order statistics and a propensity score of $s_i^g$ by the generated order statistics corresponding to each intrinsic characteristic. In this paper, we consider the following hierarchical structure of the intrinsic characteristics:

- **Customer**
    - Gender : Female & Male
    - Tenure : High-tenured (more than 5 years) & Medium-tenured (between 2 and 5 years)
    - Purchase volume : High Purchasers & Average Purchasers
- **Price**
- **Seasonal Demand**
    - Summer (May, June, July, August)
    - Winter (November, December, January, February)

The propensity scores are computed for each of the leaf nodes in the above structure, e.g., female, male, high-tenured, medium-tenured, price etc. The details about computing the scores are provided

in Section 7.2.3, 7.2.4 and 7.2.5. The propensity scores enforce an ordering of all the products with respect to both the ground-truth orders and the generated orders. We determine an agreement between these orderings using the following approach. For any two products $p_i$ and $p_j$, if $s_i^t \geq s_j^t$ and $s_i^g \geq s_j^g$ or $s_i^t < s_j^t$ and $s_i^g < s_j^g$, we count that as an agreement between the ground-truth orders and the generated orders. We randomly sample $N$ (10000 in this paper) pairs of products, find the percentage of the times the generated data and ground-truth data are concordant with each other and define that as the Relative Similarity Measure or RSM. A weak baseline, where the scores are assigned randomly to each product will produce an RSM score of 50%.

### 7.2.3 CUSTOMER CHARACTERIZATION

When a new product is launched, any e-commerce company will be interested in various characterization of the customers, who are likely to buy this product. The characterization could be based on gender, age, tenure, purchase volume etc. Once we know the characteristics of the customers, we could target relevant customers with specific deals and recommendations involving this particular product. Using ec$^2$GAN, we could generate a set of orders given a product and use the corresponding customer components for various characteristic analysis. In this paper, we analyze the gender, tenure and purchase volume of prospective customers of a product. We follow the steps below to obtain the propensity score corresponding to a customer related intrinsic characteristic:

1. First, using internal survey and historical data we train classifiers which can predict the gender, purchase volume and tenure of a customer, given its representation. These classifiers are used to map each customer to one of the categories corresponding to each characteristic. For example, corresponding to gender, the customers are mapped to either female or male.
2. For a product $p_i$, we obtain the historical purchase history for a year, i.e., we know the set of customers (say, $\{C_i\}$) who purchased this product over the last one year. We take the female (gender) characteristic to explain the method of obtaining a propensity score. The customers $\{C_i\}$ are fed to the trained gender classifier (in Step 1), to determine the fraction of female customers who purchased a product. This gives us the ground-truth propensity score $s_i^t$ for the i$^{th}$ product and for the female characteristic.
3. Each product $p_i$'s representation is also fed to the proposed ec$^2$GAN to generate 1000 fake customers (say, $\{\tilde{C}_i\}$). The generated customers are also fed to the trained classifiers (in Step 1), to determine the fraction of female customers who purchased the product. This provides us the generated orders based propensity score $s_i^g$ for the i$^{th}$ product and for the female characteristic.
4. The propensity scores $s_i^t$ and $s_i^g$ from the ground-truth data and the generated data respectively are used to compute an RSM score corresponding to the female characteristic.
5. We perform this evaluation for all of the customer related characteristics and report the results in Table 1. We obtain the RSM scores of $81.08\%$ and $82.20\%$ for the female and male characteristic respectively. Similarly, with average purchase volume, high purchase volume, medium-tenured and high tenured, ec$^2$GAN agrees with the ground truth 83.94%, 87.25%, 60.58% and 73.64% of the times respectively. These results demonstrate that the proposed ec$^2$GAN is effective in generating orders (customers in this case), which are quite similar to the real orders placed on an e-commerce website.

Table 1: Customer Characterization (RSM scores reported).

| Intrinsic Characteristics | | ec$^2$GAN | C-VAE |
|---|---|---|---|
| Gender | Female | **81.08%** | 66.84% |
| | Male | **82.20%** | 65.52% |
| Purchase Volume | Average purchasers | **83.94%** | 64.42% |
| | High purchasers | **87.25%** | 64.95% |
| Tenure | Medium-tenured | 60.58% | **66.67%** |
| | High-tenured | **73.64%** | 65.7% |

### 7.2.4 PRICE CHARACTERIZATION

When a new product is launched in an e-commerce website, accurate estimation of the price is important. The estimated price can be used to guide the sellers who might not always be aware of the actual worth of a product. Also, inaccurate catalog prices should be detected to avoid customer dissatisfaction. We can use ec$^2$GAN to estimate the price of a product and compare that with the

ground-truth price. We find all the orders corresponding to a product over the last one year. The prices are averaged to obtain the ground-truth propensity score $s_i^t$. A product vector along with various noise vectors are fed to the proposed ec$^2$GAN to generate plausible orders of the input product. We generate 1000 orders corresponding to each product. The price component of the order vectors are extracted and averaged to determine the propensity score $s_i^g$. This score indicates the propensity of a product to be of higher price. The propensity scores are used to compute an RSM value corresponding to the price and the results are reported in Table 2.

Table 2: Price Characterization (RSM scores reported).

| Characteristic | ec$^2$GAN | C-VAE |
|:---:|:---:|:---:|
| Price | 81.39% | **86.51%** |

### 7.2.5 SEASONAL DEMAND CHARACTERIZATION

For a new e-commerce product, understanding its demand across seasons is critical for stock-keeping and inventory management. Using ec$^2$GAN, we look at the seasonal distributions of the generated orders. Specifically, for a particular product, we look at the propensity scores of it being sold in the summer season (May, June, July, August) and the winter season (November, December, January, February). For each product, we look at the one year purchase history and determine the propensity scores ($s_i^t$) corresponding to the summer and winter season. For each product, we also generate 1000 orders and based on the distribution of the months, we assign a summer and winter propensity score ($s_i^g$) to the product. The propensity scores are used to compute the RSM values corresponding to the summer and winter season (Table 3).

Table 3: Seasonal Demand Characterization (RSM scores reported).

| Intrinsic Characteristics | ec$^2$GAN | C-VAE |
|:---:|:---:|:---:|
| Summer | **71.43%** | 71.28% |
| Winter | **72.22%** | 66.37% |

### 7.2.6 DISCUSSION

In this section, we discuss the results obtained in Section 7.2.3-7.2.5. We compare the proposed ec$^2$GAN with the baseline C-VAE (Section 7.2.1) for nine different use-cases (Table 1, 2 and 3). The proposed method is better than the baseline in seven out of the nine use-cases. The range of absolute improvement varied from minimum of 0.15% (summer demand prediction) to maximum of 22% (high-purchasing customer prediction). Overall, it is clear that the proposed approach is significantly better than the baseline approach. In general, we observe that the distribution of orders generated by the baseline C-VAE are peaky in nature, i.e., the variance is low. As a result of this, the generated orders often fail to capture the full spectrum of plausible orders corresponding to a particular product. However, ec$^2$GAN is much more effective in capturing the plausible order distribution of a particular product, which drives the better performance in customer characterization and seasonal demand prediction. We observe that, for price prediction, C-VAE performs better than ec$^2$GAN. Perhaps this can be attributed to the fact that the price distribution of most products have typically low variance, as the same product won't be sold at a wide-range of prices in an e-commerce website.

