# OpenReview forum: "eCommerceGAN: A Generative Adversarial Network for e-commerce"
_ICLR.cc/2018/Workshop — Accept_

### Official Review · AnonReviewer2 · 2018-02-22
**An interesting application of GANs**

**Rating:** 7
**Confidence:** 4

**Review:**

This paper proposes a GAN architecture to generate transactions in e-commerce. Each transaction is fully described by a representation of the customer, product, price, and date that are involved.

The paper is interesting and I recommend acceptance. I also have some questions:

+ Why is the representation of the product chosen as an average of the word embeddings of the product name? Would it be possible to learn the embedding vector of each product? Also, is this a pre-fitted word embedding?

+ I didn't understand why you need to add a reconstruction loss in the ec2GAN. Can you give some motivation?

+ The comment about the ecGAN ignoring outliers (Fig. 1c) seems a bit fishy to me. If it learns the true data density, then the area of outliers should have small but positive probability, such that on average we should expect as many "outliers" in that area as the proportion in the original dataset.

+ I didn't understand how the ec2GAN can be used to predict the purchase volume of a new product. I see it is straightforward to characterize the customers from generated data, but how can it be used to determine the demand?

---

### Official Review · AnonReviewer3 · 2018-03-05
**Fairly standard GAN paper, some novelty in the application but experiments don't quite show the usefulness of the system**

**Rating:** 5
**Confidence:** 4

**Review:**

The paper seeks to generate plausible e-commerce orders using GANs. This in turn involves coming up with a good representation of orders, and developing conditional GAN frameworks that allow order sequences to be conditioned on a particular product.

Using GANs for problems in e-commerce is a hot topic, though I felt the authors struggled to motivate the usefulness of such a system. Most of the tasks they describe seem like they could be addressed via traditional predictive models rather than really requiring the generative aspect.

Other than that, the paper follows pretty standard GAN methodology. The experiments are okay but a bit limited, even by the standards of GAN papers: they show some (simple) visualizations of the generated data, and that some simple statistics of the generated data match those of the real data. Each of these are valid, but again don't really show the usefulness of such a model, or that it works better than other generative alternatives (GANs are not the only generative model!).

---

### Official Review · AnonReviewer1 · 2018-03-10
**Interesting attempt for large-scale e-commerce**

**Rating:** 7
**Confidence:** 4

**Review:**

This paper proposes a GAN for e-commerce data. By employing GAN structure for
density modeling, it can faithfully mimic actual distribution of purchases by
users. to enable exploration of future items by simulating possible customer
behaviors.

The idea is good, and it works better than VAE because of its GAN structure.
The key component is that the proposed GAN would mimic possible purchase
behaviors; they show in Figure 2 that the distribution of fake sessions are
quite similar to actual purchases. While the authors provide some qualitative
analysis of actual and fake distributions in the Appendix, I would like to see
some statistical test or its Bayesian alternative here to show that these two
distributions can be regarded as the same.
Furthermore, I also would like to see that what kind of analysis are impossible
without generating fake sessions.

---

### Decision · Program_Chairs · 2018-03-20
**ICLR 2018 Workshop Acceptance Decision**

**Decision:**

Accept

**Comment:**

Congratulations, your paper was accepted to the ICLR workshop.